# Phyllotaxis Turns Over a New Leaf—A New Hypothesis

**DOI:** 10.3390/ijms21031145

**Published:** 2020-02-09

**Authors:** Derek T. A. Lamport, Li Tan, Michael Held, Marcia J. Kieliszewski

**Affiliations:** 1School of Life Sciences, University of Sussex, Falmer, Brighton BN1 9QG, UK; 2Complex Carbohydrate Research Center, University of Georgia, Athens, GA 30602, USA; tan@ccrc.uga.edu; 3Department of Chemistry and Biochemistry, Ohio University, Athens, OH 45701, USA; held@ohio.edu (M.H.); kielisze@ohio.edu (M.J.K.)

**Keywords:** Phyllotaxis, arabinogalactan proteins, auxin, calcium signaling, acid growth, Hechtian oscillator, plasma membrane ion fluxes

## Abstract

Phyllotaxis describes the periodic arrangement of plant organs most conspicuously floral. Oscillators generally underlie periodic phenomena. A hypothetical algorithm generates phyllotaxis regulated by the Hechtian growth oscillator of the stem apical meristem (SAM) protoderm. The oscillator integrates biochemical and mechanical force that regulate morphogenetic gradients of three ionic species, auxin, protons and Ca^2+^. Hechtian adhesion between cell wall and plasma membrane transduces wall stress that opens Ca^2+^ channels and reorients auxin efflux “PIN” proteins; they control the auxin-activated proton pump that dissociates Ca^2+^ bound by periplasmic arabinogalactan proteins (AGP-Ca^2+^) hence the source of cytosolic Ca^2+^ waves that activate exocytosis of wall precursors, AGPs and PIN proteins essential for morphogenesis. This novel approach identifies the critical determinants of an algorithm that generates phyllotaxis spiral and Fibonaccian symmetry: these determinants in order of their relative contribution are: (1) size of the apical meristem and the AGP-Ca^2+^ capacitor; (2) proton pump activity; (3) auxin efflux proteins; (4) Ca^2+^ channel activity; (5) Hechtian adhesion that mediates the cell wall stress vector. Arguably, AGPs and the AGP-Ca^2+^ capacitor plays a decisive role in phyllotaxis periodicity and its evolutionary origins.

## 1. Introduction

Agnes Arber [1] in “The Natural Philosophy of Plant Form” comprehensively described the development of plant morphology from the ancient philosophers—Plato, Aristotle and Theophrastus—to the more recent Cambridge botanical tradition that extends from William Turner, Nehemiah Grew and “Robin” Hill to the present. William Turner (1508–68), father of English botany. published the first herbal in English (1551) as a Fellow of Pembroke College; Nehemiah Grew (1641–1712), another Pembroke graduate, father of plant anatomy published “The Anatomy of Plants” (1682) depicted in the exquisite stained-glass windows of the college library. Finally, the Hill reaction demonstrated the photolysis of water as the source of atmospheric oxygen and established molecular botany as a new level of scientific enquiry. Arber’s historical perspective may help resolve some long-standing problems of plant morphogenesis. Thus, Arber [1] presented “The mechanism of plant morphology” and an insightful approach to the pivotal role of the cell wall and the stress–strain of cell expansion that results in *“form conditioned by pressure”* where *“even a minor [cell wall] alteration may be associated with striking changes in the external form.”* In Northcote’s laboratory, those ideas catalyzed the first Ph.D. dissertation devoted to the primary cell wall and the discovery of cell wall proteins as a new field of study. These hydroxyproline-rich glycoproteins, especially the arabinogalactan proteins (AGPs), are involved in a hypothetical Hechtian growth oscillator. It involves transduction of the wall stress–strain to the plasma membrane where an auxin-activated proton pump dissociates AGP-Ca^2+^. Elevated cytosolic Ca^2+^-activates exocytosis thus regulating plant growth. Discussion of the Hechtian Oscillator vis-a-vis the role of the primary cell wall in plant morphogenesis [2] suggests extrapolating the oscillator to phyllotaxis based on the premise that presence of the oscillator components implies the presence of a functional Hechtian Oscillator. Indeed, recent work suggests the mechanotransduction of stress relocates auxin efflux PIN proteins that generate new protoderm primordia. However, the precise biochemical mechanisms involved in stress transduction and the role of auxin and calcium homeostasis remain to be elucidated. Here, we invoke Hechtian adhesion and AGPs as essential components that lead us to propose a novel biochemical algorithm for floral phyllotaxis and an explanation of its strong tendency towards a periodic series first described by Fibonacci (1170–1240). This approach contrasts with many previous studies with an overwhelming mathematical bias. Indeed, many observations in Nature involve periodicity and the probable underlying oscillations

Oscillatory plant growth, known since Darwin [3], was subsequently confirmed by rapid tip growth of pollen tubes and root hairs [4]. Plant morphogenesis also involves periodicity strikingly displayed by the pattern of leaves and floral organs [5] that often appear as Fibonacci spirals typified by whorls of 3, 5, 8, 13, 21 and 34 petals [6]. Hypothetically, such periodicity depends on an underlying oscillator such as the recently formulated Hechtian growth oscillator [2,7] that involves auxin-driven Ca^2+^ release from arabinogalactan proteins (AGPs) of the cell surface; this hypothesis accounts for the origin of oscillations in molecular detail absent from previous models of tip growth [8]. Here, we extrapolate the Hechtian Oscillator to the challenging problem of phyllotaxis and the generation of primordia in the protoderm, the outermost cell layer of the stem apical meristem SAM. Earlier work emphasized physical factors and a mathematical approach was comprehensively reviewed in [6,9,10]. However, more recent work emphasizes a cell wall stress vector generated by rapid cell expansion in the protoderm that re-orientates auxin efflux PIN proteins of neighboring cells and thus directs auxin transport (and the inferred generation of Ca^2+^ waves) that regulate growth and differentiation (e.g., [11,12,13,14]). The present paper complements these and more recent models of [15] but with the notable exception of [16]; none consider a possible role for cell surface AGPs. However, *“Nature keeps some of her secrets longer than others”* [17]. That includes the elusive molecular function of classical AGPs [18,19,20]. Identified some fifty years ago [21,22,23], AGPs remained “A Great Puzzle” until the recent demonstration that AGP glycomodules bind Ca^2+^ specifically [24]. They form a cell surface AGP-Ca^2+^ capacitor that involves the interaction of three essential ions auxin, H^+^ and Ca^2+^. These “morphogens” of the Ca^2+^ signal transduction pathway (Figure 1) interact and thus regulate cell expansion and growth.

The pathway begins with the transduction of the cell wall stress vector to the plasma membrane, via AGP57C [25] as the likely molecular basis of Hechtian adhesion between the cell wall and the plasma membrane. Further transmission of a biochemical signal to the cytoplasm involves stretch-activated proton and Ca^2+^ ion fluxes of the plasma membrane generated by the Hechtian growth oscillator [7]. The cytoplasmic response to Ca^2+^ influx presumably involves exocytosis of wall plasticizers and precursors including redirection/reorientation of auxin efflux PIN proteins, eponymously named after their mutant pin-shaped phenotype. These auxin transport proteins channel auxin flow away from slow expansion towards rapid expansion thus generating auxin waves with maxima corresponding to the periodicity of nascent primordia. Turing’s classic paper [26] postulated only two morphogens sufficed to generate spiral phyllotactic periodicity. The sections below expand on Turing’s original suggestion with recent experimental evidence. Turing’s insight was much closer to reality than the “two interacting morphogens” he envisaged. 

The ingenuity of Mother Nature exceeds our human imagination by involving three interacting ions, auxin, protons and Ca^2+^ (Figure 1) as the master regulator of plant growth. Although ion accumulation studied for more than 80 years [27] has generally assumed the relative immobility of Ca^2+^ ionically bound to the cell wall, non-intuitively Ca^2+^ bound by cell surface AGPs now appears to be the major source of dynamic cytosolic Ca^2+^. Counter-intuitively, the mechanism for the release of dynamic Ca^2+^ from ionically bound AGP-Ca^2+^ is not obvious. However, the paired glucuronic carboxyls of AGP glycomodules explain the remarkable stoichiometric Ca^2+^-binding properties of periplasmic AGP-Ca^2+^; its dissociation by an auxin-activated proton pump predicts an essential role of AGPs in Ca^2+^ homeostasis [24].

## 2. Hechtian Adhesion

The Profound implications of Hecht [28] and many other’s observations are becoming clear. Numerous papers emphasize Hechtian strands of plasmolyzed cells but ignore the corollary, strong adhesion between the wall and plasma membrane of turgid cells which until recently has remained a scientific mystery. However, in plasmolyzed pollen tubes [7] and root hair tips [29] (Figure 2), a high density of Hechtian strands correlates rapid tip growth with Hechtian adhesion arguably mediated by AGP57C [25]. This suggests its essential role in transduction of the wall stress vector that initiates oscillations and cytosolic Ca^2+^ waves as hypothesized by the Hechtian Oscillator (Figure 1) [7].

## 3. Is the Hechtian Oscillator Just an Hypothesis? Direct Evidence

The correlation between Hechtian adhesion and tip growth also implies that transduction of the wall stress vector with concomitant activation of the proton pump releases Ca^2+^ from a tip-localized AGP-Ca^2+^ capacitor, hence a source of the tip-focused Ca^2+^ influx. Although initially an inference, direct experimental evidence was described most recently by De Vriese et al. [30]: Tobacco BY-2 cells expressing the bioluminescent Ca^2+^ sensor aequorin responded immediately to addition of the auxin analog 2,4-D, “the luminescent signal rapidly increased and reached a maximum after 90 s.” Thus, direct evidence confirms a major prediction of the Hechtian Oscillator hypothesis that connects activation of the proton pump and proton extrusion with rapidly increased cytosolic Ca^2+^ (Figure 1).

The Hechtian Oscillator exemplifies the pollen tube paradigm of rapid tip growth in particular [2,7]. The rapidly growing cell wall transmits its stress–strain status via Hechtian adhesion to the plasma membrane. The role of Hechtian adhesion in stress transduction, inexplicably overlooked for more than a hundred years, also explains how a periplasmic AGP-Ca^2+^capacitor, as a major component of the oscillator and its auxin-activated proton pump, can regulate plant growth in general. The biochemical physiological and ecological properties of the Hechtian Oscillator also avoid the vagaries of a variable external Ca^2+^ supply; it guarantees immediate access to Ca^2+^ while recycling cytosolic Ca^2+^ replenishes the AGP-Ca^2+^ capacitor. Such efficient use of Ca^2+^ may ensure the survival of calcifuge species in Ca^2+^-deficient habitats where over-expression of AGPs also observed under salt stress [31] may enhance the ability to scavenge Ca^2+^. Marine plants such as Zostera (Eelgrass) support that hypothesis. Recent characterization of their AGPs shows an elevated glucuronic acid content suggestive of enhanced Ca^2+^ binding in high salt [32]. 

## 4. Auxin Activity Is a Proxy for the Hechtian Oscillator

Heisler et al. [13] concluded that in the shoot apical meristem of Arabidopsis, *“cycles of auxin build-up and depletion accompany, and may direct, different stages of primordium development,”* further confirmed by a more recent study of Arabidopsis embryogenesis [33]. Auxin waves indicate the presence of an auxin-activated proton pump an essential component of the Hechtian Oscillator. Therefore, auxin activity itself can be taken as a proxy for an active Hechtian Oscillator, consistent with the well-known association of auxin with cell expansion.

Thus, H^+^ dissociation of periplasmic AGP-Ca^2+^ [24] is the inferred source of cytosolic Ca^2+^ that activates exocytosis in the AGP-rich protoderm. Indeed, ubiquitous distribution of AGPs throughout the Plant Kingdom [34,35] implies an absolute requirement for AGPs. Lethal knockouts of genes encoding pollen AGPs [36] confirm their essential global role. Indeed, AGPs are closely associated with morphogenesis even at the very earliest stages such as microspore embryogenesis [37]. Therefore we hypothesize that a biological oscillator generates oscillatory growth and contributes to primordia periodicity; phyllotaxis is a test case of the Hechtian Oscillator and its general applicability is developed in the following sections.

## 5. A Molecular Pin-Ball Machine Regulates Ion Fluxes at the Plasma Membrane

Auxin activates plasma membrane H^+^-ATPase proton pump by increasing its phosphorylation which increases the rate of proton extrusion [38]. The extent of ATPase phosphorylation [39] exerts fine gain control of the proton pump over a wide range. Hydrolysis of a single ATP molecule fuels the extrusion of about three protons by an H^+^-ATPase “turbo-molecular” proton pump. The molecular pathway involves successive glutamate protonation and deprotonation from the cytoplasmic side to the periplasm and cell wall (Figure 3) [39] and initial extremely fast lateral proton diffusion on the plasma membrane surface [40].

Proton extrusion and a concomitantly low wall pH associated with cell extension so-called “acid growth” [41] exemplify Lord Rutherford’s dictum that *“No experimental result is ever wrong.”* A widely accepted (textbook) explanation invokes low pH-dependent wall loosening “enzymes” and expansins all of unknown specificity [42] but ignores the dissociation of AGP-Ca^2+^ that provides an alternative reinterpretation of “acid growth” based on a visual analogy of the plasma membrane depicted as a metaphorical “molecular pin-ball machine” in (Appendix A) that regulates three ion fluxes, H^+^, Ca^2+^ and auxin (anions at neutral pH, neutral at low pH). When activated by auxin, the proton pump shoots fast protons into the periplasm where they dislodge Ca^2+^ ions from the periplasmic AGP glycomodules; i.e., proton efflux generates Ca^2+^ influx. Thus, free Ca^2+^ ions then enter the cytosol via stretch-activated Ca^2+^channels. The rapid increase of cytosolic Ca^2+^ [30] activates exocytosis of putative wall plasticizers, namely AGPs and AGP peptides [7] (Figure 1). Regulation of Ca^2+^ homeostasis is thus the major function of the proton pump rather than the regulation of wall pH.

## 6. Transduction of the Stress Vector through the Protoderm

Primordia are initiated in a “generative annulus” [6]. This thin band of protodermal cells encircles the outermost cell layer of the stem apical meristem (SAM) protoderm where its powerful morphogenetic properties are triggered by the incipient primordia. These most rapidly expanding cells transmit the stress vector to neighboring cells via their anticlinal walls [14,43,44,45]. Such stress relocates the auxin efflux “PIN” proteins so their polarization in the protoderm results in auxin transport towards incipient primordia. The physical basis of stress transduction depends on Hechtian adhesion while its co-localized auxin efflux “PIN” proteins were shown by immunocytochemistry [45]. Such polarized localization of auxin efflux proteins, e.g., PIN1 in the protoderm [33], suggested a crucial role for auxin transport in the generation of primordia [46]. It was concluded that *“PIN directs auxin to the sites where young primordia are being formed.”* Rapid relocation of PIN1 is evidently of huge significance, although the precise mechanism remains obscure. Two components not previously considered essential for primordia formation involve Hechtian adhesion and AGPs. They mediate transduction of the cell wall stress vector, as follows:

## 7. Stress, PIN Protein Redirection and Auxin Waves

*“Symmetries control distribution in space”* [47] begs the question: What is the origin of symmetry, and how is it broken? This centuries’ old debate gradually developed from “vital force” and the equally unfalsifiable “morphic resonance” to Spemann’s organizer, morphogenetic fields, Waddington’s “evocators” and Turing’s morphogen gradients to current concepts of homeobox genes and a plethora of cognate transcription factors. They illustrate the complexity of animal morphogenesis compared with the sublime sessile simplicity of plants and the view here that auxin gradients control proton and Ca^2+^ fluxes that are predominant regulators of growth and differentiation. 

Auxin transport involves diffusion facilitated by auxin efflux, eponymous “PIN” proteins essential to the generation of auxin waves that break the perfect symmetry of the protoderm as follows: 

Morphogenesis frequently involves auxin waves [48]. That includes phyllotaxis [5] where a recent theoretical biophysical model involving complex linear wave equations predicts auxin waves that specify the site of new primordia [16]. Those authors noted that: *“The role of auxin transport in phyllotaxis must be universal”* and also inferred that *“electromechanical feedbacks apparently involve the Ca^2+^ and H^+^ ions.”* Recent direct experimental evidence [7,24] explains how the cell wall stress vector and transcytosis relocate PIN proteins and thus together with AGP-Ca^2+^ generate the auxin and Ca^2+^ waves that initiate primordia formation.

Rapidly expanding cells of the protoderm transmit the stress vector via anticlinal walls towards slower cell expansion (Figure 4). The biophysical basis of stress transduction arguably involves Hechtian adhesion between the wall and plasma membrane as described for the growth of pollen tubes and epidermal cells of root tips [45]. Hechtian adhesion is also evident in the protoderm: for example, Figure 1A of [49] shows a Hechtian strand formation in protodermal areas. Hechtian adhesion is virtually universal and thus also present in actively growing tissue like the protoderm! Furthermore, auxin waves are also evidence of an active Hechtian Oscillator based on Hechtian adhesion.

Transmission of the stress vector relocates PIN auxin efflux proteins to the stressed anticlinal walls of stressed cells [45]. Thus, auxin moves against its concentration gradient towards cells with the highest auxin levels therefore depleting the auxin of less rapidly expanding cells. Channeling auxin towards stress generators, i.e., the most rapidly expanding cells, presumably initiates primordia when auxin reaches a critical threshold level [15]. Attenuation of the stress vector by intervening distal cells depleted of their auxin slows their expansion until a boundary “tipping point” of minimum cell expansion appears (Figure 4) where the stress vector reverses its direction with concomitant reversal of PIN protein polarity [45]. Thus, a new auxin gradient increases towards a new stress vector initiated by cell expansion of a newly formed primordium. Hence, auxin waves appear as peaks that coincide with new primordia separated by auxin troughs; these repeat what is in essence an autocatalytic cycle (Figure 5):

We propose that the generation of successive auxin waves varying in amplitude and frequency depends on the response of the proton pump and cell surface AGP. Indeed, there is an increasingly clear correlation between enhanced AGP expression and tissue morphogenesis. 

Membrane-bound PIN proteins recycle rapidly via transcytosis; for example, the relocalization of PIN7 occurs within two minutes after the gravity stimulus [50]. Arguably, the mechanism involves Hechtian adhesion that transmits wall stress directly to the plasma membrane rather than indirect transmission via “statoliths.” During rapid tip growth of pollen tubes and root hairs, Hechtian adhesion predominates at the growing tip where wall stress–strain is most apparent and exocytosis is maximal. This correlation suggests that the stress vector relocates Hechtian adhesion sites at a malleable cell wall and this, in turn, directs the exocytosis of wall precursors including auxin efflux PIN proteins (cf. Figure 4)**.** Thus, auxin waves and cytosolic Ca^2+^ [51] generated by transmission of the cell wall stress vector depend on two additional factors, transcytosis and cell wall rheology.

## 8. Cell Wall Rheology 

Transmission of the stress vector from rapidly expanding cells of incipient primordia involves plasticity of the anticlinal cell walls. Although Anton Heyn [52] identified wall plasticity as a crucial determinant of cell expansion, even after eighty years the biochemical basis of the Heyn paradigm remains debatable. Despite Heyn’s emphasis on plasticity, cleavage of covalent cell wall crosslinks remain the predominant but elusive explanation [53]. Most synthetic plastics depend on plasticizers like phthalates, small molecules that disrupt the alignment of linear polymer chains but do not cleave covalent bonds. Analogous plasticizers of pectins include classical AGPs but their molecular size precludes simple diffusion through a pectic matrix. However, the much smaller diffusible AGP peptides upregulated by auxin [54] are also potential plasticizers; significantly their glucuronic acid content [55] indicates Ca^2+^-binding capacity similar to the much larger non-diffusible classical AGPs typified by LeAGP1 that transit the wall by extrusion rather than diffusion [31]. Thus small AGP peptides diffusing though the wall can compete for Ca^2+^ crosslinks and thus favor a pectic gel-sol transition with a concomitant increased wall plasticity. Pectin methyl esterases have also been invoked [56]; Cosgrove [57] notes *“localized deesterification of homogalacturonan as a signature event in the auxin-induced patterning of the shoot apical meristem… this correlation of de-esterified pectin with softer meristem regions is perplexing”* but consistent with electrostatic repulsion of ionized pectic carboxylates their Ca^2+^ depleted and scavenged by AGPs and AGP peptides with a higher affinity for Ca^2+^. However, Altartouri et al. [58] represent the prevailing view that Ca^2+^ crosslinkage of de-esterified pectin decreases wall plasticity. This implicitly assumes sufficient free Ca^2+^ for pectin crosslinking but ignores AGP-Ca^2+^ homeostasis that determines the availability of both free and bound apoplastic Ca^2+^.

Fine control of pectin rheology by small diffusible AGP peptides has not previously been considered. Similar reasoning may apply to some monocots where glucuronoxylans largely replace pectin [59]. Finally, we can only agree that: *“Cell expansion thus appears to be intimately linked to these wall sensor pathways in ways we are only beginning to fathom.”* [57].

## 9. A Phyllotaxis Algorithm 

*“While progress has been made, there are many fascinating challenges in phyllotaxis still open for the curious mind to explore. The story is far from over. While careful experiments are crucial to continued progress, it does not require elaborate experiments for ordinary folk to enjoy the wonderful architectures seen near the meristems of plants”* [6]. Summarizing much recent work: *“Key results stem from the observed facts that phyllotactic patterns are naturally produced by instabilities, connected to both the distribution of the growth hormone auxin and to the local stress–strain fields.”* Although those “instabilities” remain undefined, phyllotaxis per se is remarkably stable but with exceptions described by Arber [1] in several species. For example, a completely dimerous flower of Iris on a shoot also bearing a normal trimerous flower, and Potentilla flowers with three, four, five or six petals and concluded that “phyllotaxis depends upon the rhythmic development of primordia at the growing apex.” Somewhat ahead of its time, eighty years ago, the insightful observation was made: “it seems reasonable to suppose that these variations are associated with internal chemical oscillations” with a final intuitive leap to “Physico-chemical factors...one such factor has been so universal as to affect the whole of the Plant Kingdom; this is the development of a cell wall encasing each unit of the plant body.” “The challenge now is to describe how the stem apical meristem generates phyllotactic patterns de novo” [12]. The historical emphasis on mathematical approaches based on optimal packing shows that Fibonacci patterns can arise naturally in many pattern-forming systems but this is not obviously connected with the biochemical mechanisms involved in patterning. Both approaches achieve optimal packing but in quite different ways. All the components of the Hechtian Oscillator are present. Thus, a dynamic algorithm involving protoderm biochemistry and mechanotransduction is now feasible as a working hypothesis.

Protoderm cell expansion generates new primordia (*N*). For example in a floral phyllotaxis, *N* is a function of the stem apical meristem (SAM) size and the magnitude of major variables that define the symmetry and periodicity of new primordia. To sustain their growth, rapidly expanding cells demand auxin by generating the cell wall stress vector (CW_sv_) that redirects PIN proteins thus channeling auxin towards these incipient primordia (Figure 4). A resulting auxin gradient then appears as waves in the annulus, a narrow band of morphogenetic cells surrounding the outer protoderm (SAMP_a_) with auxin maxima and minima corresponding to future primordia and boundary tipping points, respectively. Generally, increasing the magnitude of a variable increases auxin transport towards a primordium hence rapidly depleting distal cells. Reversal of PIN protein orientation then generates a new primordium. Thus, an increased auxin depletion rate increases the number of new primordia. However, they also depend on Ca^2+^ availability determined by the expression of AGPs. We predict that AGPs strongly expressed in the protoderm will increase the periodicity of primordia while weaker expression will decrease it. Thus, to that extent, the algorithm is semi-quantitative and dependent on strong expression of AGPs in the protoderm of Arabidopsis meristems [60], Euphorbia embryonic cultures [61] and during somatic embryogenesis of Arabidopsis [62]. The novel suggestion that AGPs play a decisive role as crucial determinants of phyllotaxis periodicity (Figure 6) depends on a complex function of a hypothetical algorithm derived from the foregoing considerations:

Stem apical meristem protoderm SAMP_a_ generates ***N*** new primordia as a function of an equation comprised of the following variables: AGP-Ca^2+^ capacitor: AGP_c_;Stem apical meristem protoderm annulus radius: SAMP_a_;PP proton pump activity: PP;Auxin efflux activity: A_efflux_ (hence auxin levels: A_ux_);Ca^2+^ channels: C_ch_;Exocytosis is a complex variable regulated by Ca^2+^ influx;Hechtian adhesion: H_ad_;Cell wall stress vector: CW_sv_;
$$\frac{{\mathrm{SAMP}}_{a}}{{\mathrm{AGP}}^{-1}+{\left(\mathrm{PP}\cdot A_{\mathrm{ux}}\right)}^{-1}+(C_{\mathrm{ch}}{}^{-1}+CW_{\mathrm{sv}})+H_{\mathrm{ad}}{}^{-1}}$$

The cell wall stress CW_sv_ vector determines auxin, proton and Ca^2+^ ion fluxes involving four phases of the Hechtian Oscillator proton pump that regulate cell expansion.

Phase I. Quiescent: Minimal cell wall stress corresponds to minimal Ca^2+^ influx and minimal activity of the oscillator.

Phase II. Activation: Cell expansion increases wall stress, demand for auxin and opens Ca^2+^ channels; entry of Ca^2+^ initiates auxin binding to the proton pump and initial oscillator activation leading to Phase III.

Phase III. Maximum Activation: Occurs at the high auxin levels supplied by redirected auxin efflux PIN proteins; accelerated proton extrusion dissociates glycomodule AGP-Ca^2+^and supplies the Ca^2+^ channels thus generating cytosolic Ca^2+^waves that activate exocytosis, notably of wall precursors and plasticizers but also enabling dynamic redirection of PIN proteins. Addition of precursors reinforce the wall and slow its expansion, leading to Phase IV:

Phase IV. Return to quiescent phase: Stress relaxation of reinforced wall closes Ca^2+^ channels.

Cytosolic Ca^2+^ recycles via AGP precursors and Ca^2+^transporters replenish periplasmic AGP-Ca^2+^

When attenuation of the stress vector reaches a tipping point of minimal oscillator activity and least rapid cell expansion, distant cells expand more rapidly and now exert a new stress vector in an opposing direction thus generating a new primordium that contributes to phyllotactic symmetry.

SAMP_a_ stem apical meristem protoderm generates new primordia as a function of the activity of five major variables or determinants shown in the speculative graph taking petal phyllotaxis as an example ranging from 3 to 34 petals. It assumes that the cell wall stress vector CW_sv_ is essentially constant for a given SAMP_a_. 

Protoderm AGP content (activity) is the primary determinant based on the simple hypothesis that size of the AGP-Ca^2+^ capacitor determines the periodicity of primordia initiation (cf. Figure 4) where a large capacitor generates more primordia. The graph correlates the Fibonacci series with AGP expression and activity of other factors particularly the proton pump not previously connected with phyllotaxis. Other determinants illustrate an inferred hierarchy based on the size of their relative input to the Hechtian growth oscillator.

The beguiling simplicity of the above plot inferred largely from biochemical observations is in strong contrast to previous complex formulations based largely on mathematical/geometrical logic.

## 10. Evolutionary Origin of Angiosperm Phyllotaxis

The evolutionary history of the stem apical meristem from a relatively simple arrangement of apical cells in the Bryophytes and ferns culminates in the morphogenetic protoderm of the angiosperms. Here, we conjecture that hybridization may solve the riddle of Fibonacci phyllotaxis and its evolutionary origins. 

Strong AGP expression predicts numerous closely spaced primordia (Figure 4) that generate floral phyllotaxis, arguably determined by the amplitude of the Ca^2+^ signal that depends on the AGP-Ca^2+^ capacitor size, proton flux and Ca^2+^ channel status. Thus, cells of the protoderm with a large AGP-Ca^2+^ capacitor will increase cytosolic Ca^2+^ rapidly and therefore generate numerous primordia within a shorter range than in a protoderm with weaker AGP expression and therefore with more cells between primordia. For example, lower activity of the proton pump and a smaller AGP-Ca^2+^ capacitor increase spacing between primordia. Unlike the animal kingdom, plants have an enormous propensity for polyploidy and hybridization that we suggest provides a simple biochemical explanation for an evolutionary origin of the well-known Fibonacci series of floral organs exemplified by the 3, 5, 8, 13, 21 and 34 petals (Figure 6), e.g., *Ranunculus ficaria* (13), *Erigeron canadensis* (21). Can hybridization between contiguous members of the series generate a Fibonacci sequence? If so, how? Consider a hybrid expressing the sum of AGPs from both parents! If AGPs play a dominant role in defining phyllotaxis, then a hybrid of two-fold and threefold symmetry, with a corresponding increased size of the AGP-Ca^2+^ capacitor, would generate the most common five-fold symmetry, and so on for subsequent members of the series (Figure 6). Thompson [17] viewed the Fibonacci series not as the cause but merely *“a consequence of optimal space filling in systems adding new units at a pole.”* Thus, we infer that hybridization generates floral Fibonacci phyllotaxis and accounts for the evolutionary origin of a discrete series rather than a smooth arithmetic progression. This suggestion has the merit of simplicity based on Occam’s Razor in contrast to all preceding mathematical conjectures [9] and is supported by the Hechtian Oscillator as a predictive paradigm.

## 11. D’Arcy Thompson, Alan Turing and Peter Mitchell Revisited

Thompson’s classical “Growth and Form” [17] exemplified a purely descriptive mathematical approach that collated a huge corpus of biophysical observations rather than hypothesis-driven experiments. On the other hand, Turing [26] combined a mathematical with a physical-chemical approach. Thus, while a Turing self-replicating machine aptly fits cell replication, Turing postulated that ontogeny based on biological parsimony might involve the diffusion of only two chemical morphogens that would suffice to create morphogenetic gradients. Those ideas preceded the biochemical insights of Mitchell (Mitchell [63], experimentalist par excellence who questioned conventional wisdom and proposed the versatile chemiosmotic proton pump). A universal energy transduction machine couples proton gradients across lipid membranes to generate ATP that energizes all life. In reverse, it consumes ATP and pumps protons. This vectorial chemical system differs fundamentally from conventionally scalar chemical ones as explained by Mitchell [64]: *“It was obviously my hope that the chemiosmotic rationale of vectorial metabolism and biological energy transfer might one day come to be generally accepted, and I have done my best to argue in favour of that state of affairs for more than twenty years…was it not the great Max Planck who remarked that a new scientific idea does not triumph by convincing its opponents, but rather because its opponents eventually die?”* Although Mitchell’s unconventional ideas were initially rejected, they were finally recognized. Their universal applicability has become apparent more recently. In simple photoautotrophs, light-driven proton gradients involve bacteriorhodopsin [65], while in more advanced eukaryotes, an electron transport chain generates mitochondrial proton gradients. Proton pumps and their regulation are thus at the epicenter of plant growth that, stripped to its bare essentials, depend on three morphogen gradients, auxin, protons and Ca^2+^ rather than just two. 

However, these gradients do not arise by simple diffusion but are regulated by auxin efflux “PIN” proteins whose discovery began with Rubery and Sheldrake’s [66] classic experiments in the laboratory of Northcote [67]. PIN proteins control auxin gradients and auxin levels that activate the proton pump while the cell wall stress vector opens Ca^2+^ channels that generate cytosolic Ca^2+^ gradients. Thus, the Hechtian growth oscillator is an extrapolation of Mitchell’s chemiosmosis that unifies physics and chemistry in a minimalist approach to regulating plant growth. Indeed, precursors to life surely involve proton gradients as a basis of prebiotic energy transduction and the universal proton pump of exoplanet life in the habitable zone.

## Figures and Tables

**Figure 1 ijms-21-01145-f001:**
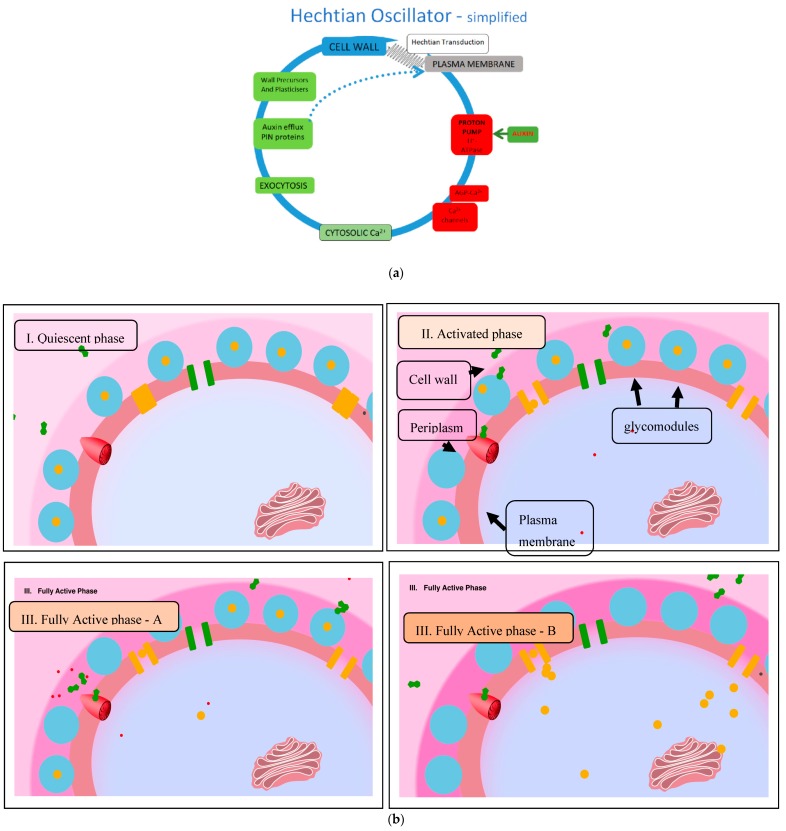
(**a**) The Hechtian Oscillator ion fluxes regulate growth. Depicts a simplified version of the Hechtian Oscillator in [2]. This figure shows stills from the animation in Appendix A. Membrane and ion fluxes are analogous to a molecular “pin-ball machine.” KEY: protons: red, Ca^2+^ ions: yellow, auxin: green, stretch-activated Ca^2+^ channels; Ca^2+^ trickle initiates proton pump activity: (**b**) **Phase I. Quiescent: [7s]** Proton pump minimally active; Ca^2+^ channels closed with minimal Ca^2+^ influx. **Phase II. Activation: [6s]** Turgor increases cell expansion and thus wall stress that increases demand for auxin and opens stretch-activated Ca^2+^ channels; Ca^2+^ trickle initiates auxin binding by the proton pump, initiating low-level oscillator activity leading to Phase III. **Phase III. Fully Activated: [12s]** high auxin levels fully activate proton pump. Proton extrusion dissociates periplasmic glycomodule AGP-Ca^2+^. Entry via Ca^2+^ channels generates cytosolic Ca^2+^ waves that activate: exocytosis of: cell wall precursors, wall plasticizers and redirect auxin efflux “PIN” proteins. **Phase IV: [9s]** Returns to Quiescent state: Stress relaxation closes Ca^2+^ channels. Auxin dissociates from proton pump; cytosolic Ca^2+^ recycles to recharge glycomodules and determine phyllotaxis periodicity as follows.

**Figure 2 ijms-21-01145-f002:**
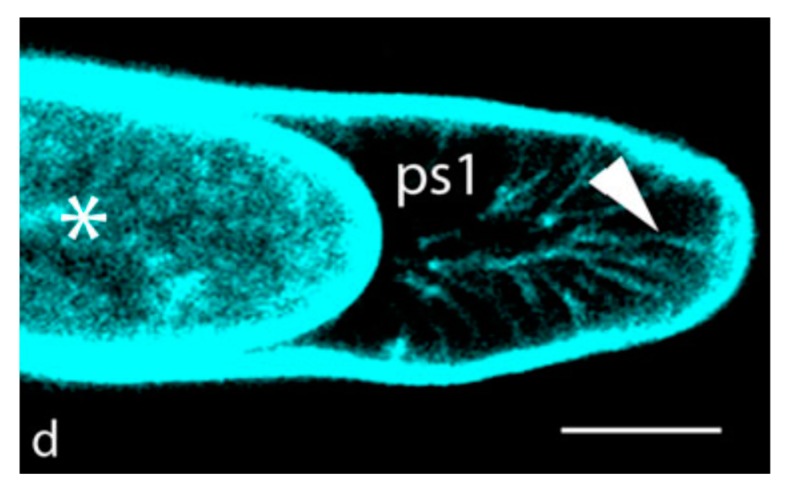
Hechtian strands in root hairs (arrow head) towards the very tip after labeling wheat root hairs with a membrane selective non-permeable fluorescent styryl dye, FM1-43. Reprinted with permission from [29].

**Figure 3 ijms-21-01145-f003:**
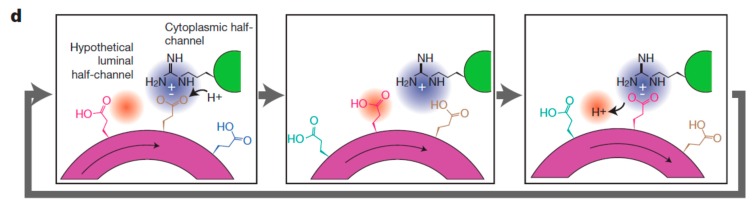
The proton pump pathway. Sequential protonation and deprotonation of the c-ring involve ATP-hydrolysis-driven rotation that causes protonation of a Glu residue at the cytoplasmic half-channel with subsequent deprotonation of a Glu residue at a luminal half-channel. Reprinted with permission from [39].

**Figure 4 ijms-21-01145-f004:**
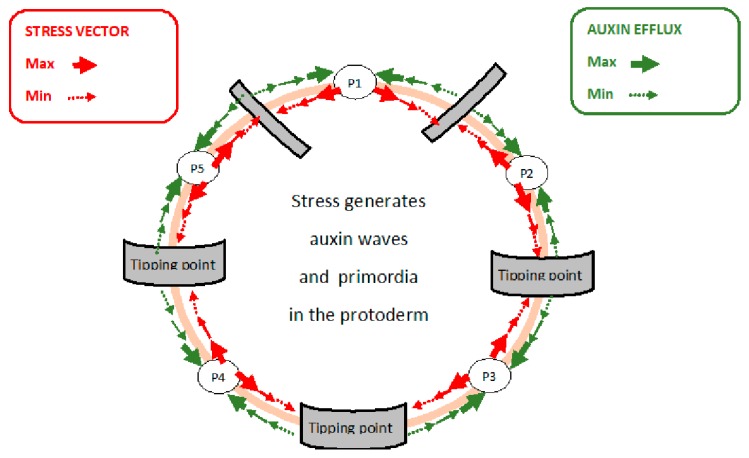
Stress in arabinogalactan proteins (AGP)-rich protoderm generates primordia. This hypothetical scheme illustrates the possible origin of auxin waves in phyllotaxis [6]. Five-fold rotational symmetry predominates as the archetype in dicot floral phyllotaxis. A plausible biochemical algorithm generates auxin waves and new primordia: Rapid cell expansion creates the stress vector (red arrows) that orientates PIN proteins; these channel auxin (green arrows) towards rapidly expanding cells to form incipient primordia and deplete auxin from sites of slow expansion until reaching a “tipping point” (lowest auxin level, slowest cell expansion, minimal stress) where PIN proteins reverse their orientation. Auxin maxima and minima generate regions of rapid expansion at auxin peaks corresponding to incipient primordia P1 to P5 separated by slowest growth at auxin troughs or “tipping points.” Precise spacing of growth peaks corresponds to the frequency of auxin waves controlled by three primary determinants, proton pump, auxin flux and AGP-Ca^2+^ capacitor size.

**Figure 5 ijms-21-01145-f005:**
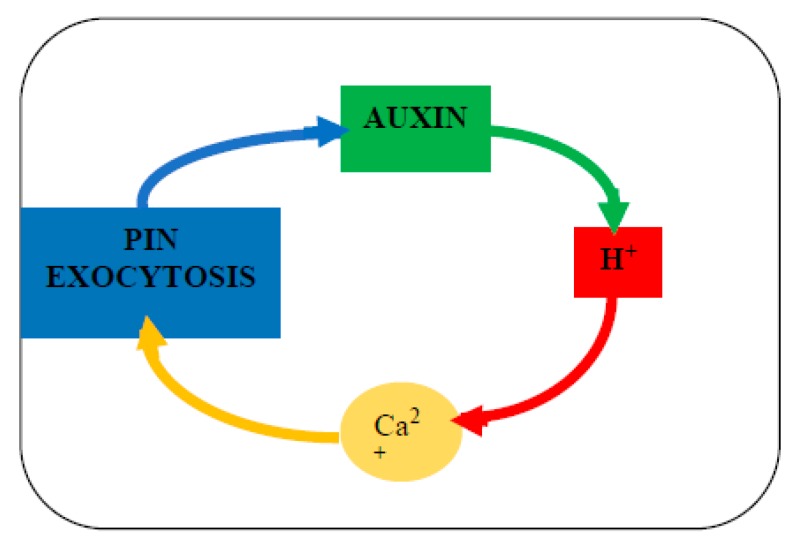
The auxin autocatalytic cycle. While Turing proposed a model based on the interaction between simple diffusion gradients of two morphogens, we suggest the interaction of three ions: PIN proteins boost the uphill diffusion of auxin against the concentration gradient while protons the fastest diffusing ions, dissociate AGP-Ca^2+^ thus enhancing exocytosis of PIN proteins (and Ca^2+^channels) that propagate the Ca^2+^ message. This is summarized by the canalization theory [12] in which *“small local differences in auxin concentration are amplified by a self-reinforcing accumulation mechanism, resulting in local auxin elevation and auxin depletion in the surrounding tissue.”*

**Figure 6 ijms-21-01145-f006:**
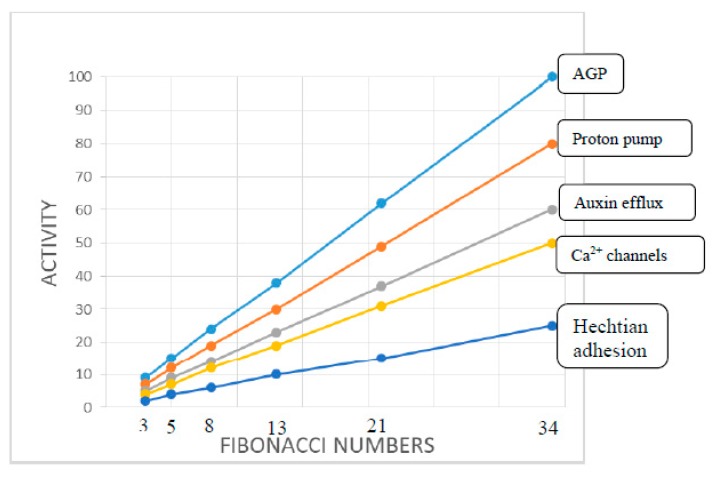
Major variables of the phyllotaxis algorithm.

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
