# Peer review of "Phyllotaxis Turns Over a New Leaf—A New Hypothesis"

_ijms, 2020, doi:10.3390/ijms21031145_

Round 1
Reviewer 1 Report
The review Phyllotaxis turn over a new leaf by Lamport et al is well written covering all the references in the field. The authors have written in detail how AGPs, AGP- Ca2+, proton pump and auxin play a decisive role in phyllotaxis periodicity. The review can be accepted without anychanges.
Author Response
Thanks for the positive review.
Minor corrections made.
Reviewer 2 Report
The authors of this review provide a timely examination about the critical determinants of an algorithm that generates phyllotaxis spiral. In light of the quest to elucidate what triggers phyllotaxis and which are the downstream signaling pathways/events that finally lead to eventually to the establishment of the phyllotaxis itself, this article contributes significantly to our understanding and provides the basis for further studies.
I therefore think that the manuscript should by published prior minor revision so as the authors to check some few liguistic mistakes.
Author Response

(The authors gave the same response as above.)

Reviewer 3 Report
I’m not convinced by the interest for International Journal of Molecular Sciences to publish the present ms ijms-645441 « Phyllotaxis turns over a new leaf ».
In my opinion this work is far from interest for IJMS.
For sure, the subject treated by the present review could be of interest, if a more detailed view of the molecular mechanisms controlling phyllotaxis is required for publishing in such a Journal to attract its audience.
63 references are cited but only 17 within the last 5 years (most of them not directly on the subject), it’s quite low, and clearly evidenced the narrowest of the treated subject.
What is new since the last review on the subject cited by the Authors (reference 6, published in 2015 in Physica D, cited only 14 times) is also unclear?
Moreover, the quality of the illustrations is very low and far from the standards required for publication.
For all these reasons I cannot recommend this work for publication in IJMS. I suggest the Authors to submit this work to a more specific journal focusing on plant development.
Author Response
Disappointingly, the third reviewer has not grasped the essential role of AGPs in membrane dynamics summarised by Mr. Ben Coleman’s beautiful animation of three ion fluxes depicted in the supplementary video. Proton, Ca2+ and auxin ion fluxes generated by the Hechtian Growth Oscillator have wide ramifications in the regulation of plant growth. Phyllotaxis provides a useful “stress test” of Hechtian hypothesis validity. Phyllotaxis is particularly relevant judging by much current interest, although the reviewer castigates me by asking “What is new” since a review in 2015, but ignored our more recent references dealing with phyllotaxis 5, 10 and 15 [published in 2019; 2016 and 2018]. Those papers together with the most recent offering from Zhao et al. (2019) Plant Physiology 181:1191-1206 made us realise that the AGP-Ca2+ connection added a significantly new perspective to the field of phyllotaxis and simply allowed us to “connect the dots” and integrate much previous work into a new conceptual landscape.
With a little more insight third reviewer would have appreciated the novel aspects of our paper summarised by the ten highlights listed below, of which two are of particular significance: Firstly, the biochemical basis of an algorithm for phyllotaxis; secondly, a truly unique description of the plant plasma membrane as a “molecular pin-ball machine” that regulates Ca2+ homeostasis and thus plant growth in general. That in itself could have merited an entire paper. It might seem a far-fetched idea but there is tangible evidence in Reference 30; De Vriese et al. (2019) report that proton pump activation results in immediate Ca2+ influx, precisely as the Hechtian Oscillator would predict.
The third reviewer raises other points as follows:
The reviewer is obviously not aware that our paper was written for the IJMS special issue on cell wall proteins in mind, and is therefore highly appropriate for an IJMS issue devoted to the field of cell wall proteins that I pioneered in 1960 with an unparalleled track record of significant discoveries in that area. The third reviewer may well be eminent in his or her field but certainly not in ours! Thus to be so dismissive of our work betrays ignorance and prejudice. To be fair, the reviewer is obviously not a biochemist, so our biochemical and biological argument may just have proved too subtle. The reviewer suggests we need a “detailed view of the molecular mechanisms controlling phyllotaxis”. One has to wonder if that reviewer read our paper or merely scanned it. Apparently we haven’t cited a sufficient number of recent papers but does not mention any egregious omissions. We listed novel aspects, highlights of our paper in some detail below. Quality of the illustrations? Of course there is always room for improvement but none of our illustrations call for detailed high resolution and are therefore probably adequate. Reviewer suggests that we should submit this work to a more specific journal focusing on plant development. However, it is intended for the special issue on cell wall proteins.AGP-Ca2+ offers a unique perspective to phyllotaxis:
AGP glycomodules act as a dynamic cell surface Ca2+ capacitorkl AGP-Ca2+ source of cytosolic Ca2+ hence Ca2+ homeostasis and signallingk AGP-Ca2+ powers the Hechtian oscillator that regulates plant growthm Mitchell’s proton pumpn generates plasma membrane ion fluxes
i.e. proton efflux generates Ca2+ influx
Auxin efflux “PIN” proteins regulate Mitchell’s proton pumpn Animationo of three ion fluxes: Auxin activates proton pump releasing bound Ca2+ Plasma Membrane is a unique “molecular PIN-ball machine!” [supplementary video]o Auxin is a proxy for the Hechtian oscillatoro A Phyllotaxis Algorithm involves the Hechtian oscillatoro Floral phyllotaxis evolved by hybridisationoTimeline of discovery by the Kieliszewski/Lamport Labs:
a.1960 Lamport & Northcote Nature – The founder event
b.1963 Lamport J.Biol. Chem. – Hydroxyproline hydroxyl requires Oxygen fixation
c.1967 Lamport Nature - Hydroxyproline-O-glycosidic linkage
d.1970 Lamport Ann. Rev. Plant Physiol. – AGP discovery
e.1999 Gao et al. Plant J. – LeAGP1 purification
f.2002 Zhao et al. Plant J. – Hyp glycosylation code
g.2003 Tan et al. Plant Physiology – peptide motifs direct glycosylation
h.2004 Tan et al. J.Biol.Chem. – glycomodule structure
i.2006 Lamport et al. New Phytologist – location quantified
j.2010 Tan et al. J.Biol.Chem. – glycomodule repetitive motif
k.2013 Lamport & Varnai New Phytologist – AGP-Ca2+ capacitor
l.2014 Lamport et al. Ann. Bot. – AGP–Ca2+flux capacitor
m.2018 Lamport et al. New Phytologist – Hechtian Oscillator hypothesis
n.2018 Lamport et al. Int.J.Mol.Sci. – AGP morphogenetic role reviewed
o.2020 Lamport et al. Int.J.Mol.Sci. – AGPs regulate phyllotaxis
Round 2
Reviewer 3 Report
First of all, no I did not observed that this work was part of a special issue on cell wall protein, which obviously clears my question of the choice of the Journal for this work. When I wrote this comment I had in mind “Development” as a better option for this work, so by reading your comments I realized that there is a clear misunderstood.
Therefore, before going into the details, I just want to clarify that the style (brief and direct) use for this review was neither arrogance nor aggressiveness towards you. My apologies if this has been interpreted that way this is really not how I behave and it was not the way this review have to be interpreted.
To go back to my review, thank you very much for your very convincing answer. In my opinion, it would be an obvious add value if this pitch or something similar that undoubtedly presents the novelty of this work, in particular since the mentioned Review published in 2015. This represents in my opinion both the justification, but more importantly the need of this new (re)view.
The choice of a non-specialist Journal, even if this work is published as a part of a special issue, must result in the use of a simpler (not simple) catchphrase at least to introduce this work in order to attract the attention of non-specialist readers, a target not to be overlooked for IJMS. Presented in this simple and convincing words, the interest of this paper is obvious ("simple is the best" ... as a communication skills of course). So, I suggest you consider this possibility. If not, I think the opinion of the other two reviewers was clear enough, and I will therefore agree with this opinion.
Author Response
Your considered reply is much appreciated and many thanks indeed!